ə | **Open Peer Review** | Virology | Methods and Protocols

# Human PrP E219K: a new and promising substrate for robust RT-QuIC amplification of human prions with potential for strain discrimination

A. Marín-Moreno,[1] F. Reine,[1] F. Jaffrézic,[2] L. Herzog,[1] H. Rezaei,[1] I. Quadrio,[3,4] S. Haïk,[5] V. Béringue,[1] D. Martin[1]

**ABSTRACT**   Mammalian prion diseases are fatal neurodegenerative disorders caused by the conformational conversion of the host-encoded prion protein (PrP) into a pathogenic, misfolded isoform, known as PrP$^{Sc}$. Definitive diagnosis currently relies on post-mortem histopathological examination of the central nervous system. Among emerging diagnostic tools, *in vitro* amplification techniques such as Real-Time Quaking-Induced Conversion (RT-QuIC) have demonstrated high sensitivity, specificity, and speed, although certain prion strains remain difficult to amplify. Here, we evaluate a novel recombinant substrate for RT-QuIC: human PrP E219K, a naturally occurring polymorphism in which lysine substitutes glutamic acid at codon 219. Using this substrate, we successfully amplified six sporadic Creutzfeldt-Jakob disease (sCJD) strains and the variant Creutzfeldt-Jakob disease (vCJD) strain from both human PrP transgenic (tg650) mouse brain homogenates and directly from patients' samples. In tg650-passaged prions, amplification reactions were initiated between 3 and 36 hours for sCJD prions and between 11 and 31 hours for vCJD prions, covering a 5- to 7-log dilution range depending on the strain. For patient brain homogenates, amplification reactions started between 0 and 27 hours for sCJDs and between 17 and 35 hours for vCJD, covering a 5- to 8-log dilution range depending on the strain. VV1 prions from a patient sample could only be amplified over a 2-log dilution range. Moreover, lag times of amplification reactions enabled reliable discrimination between vCJD and all tested sCJD subtypes. These findings represent a significant advance toward ante-mortem typing of human prion diseases.

**IMPORTANCE**   Creutzfeldt-Jakob disease (CJD) is a fatal neurodegenerative disorder within the prion disease family. Definitive diagnosis and strain typing currently require post-mortem analyses. In this study, we demonstrate that the human PrP E219K variant serves as an effective substrate for prion in vitro amplification using real-time quaking-induced conversion. This substrate enables (i) rapid and robust amplification of the most common human prion strains and (ii) clear and direct discrimination between variant CJD and all tested sporadic CJD subtypes, based on statistical analyses of lag times of the prion amplification reactions. These findings represent a significant step toward the development of ante-mortem tools for prion strain typing in affected patients.

**KEYWORDS**   prion, Creutzfeldt-Jakob disease, RT-QuIC, prion amplification, strain discrimination, strain typing

Transmissible Spongiform Encephalopathies (TSE), or prion diseases, are fatal neurodegenerative disorders that affect various mammalian species, including ruminants (such as sheep, cattle, and cervids) and humans (1–3). Experimentally, TSE can also be transmitted to rodents such as mice, hamsters, and bank voles (4). In humans,

Address correspondence to D. Martin, davy.martin@inrae.fr, or V. Béringue, vincent.beringue@inrae.fr.

The authors declare no conflict of interest.

See the funding table on p. 10.

10.1128/spectrum.00292-25 **1**

Creutzfeldt-Jakob disease (CJD) is the most common form of prion disease. Sporadic CJD (sCJD) occurs at a rate of approximately 1.5 cases per million people annually and accounts for 80% of human TSE cases (5). Other human TSEs include genetic forms, such as fatal familial insomnia and Gerstmann-Sträussler-Scheinker syndrome, and acquired forms, such as variant CJD (vCJD), which is associated with the consumption of food contaminated by bovine prions responsible for bovine spongiform encephalopathy (6).

Prion diseases result from the conformational conversion of the host cellular prion protein ($PrP^C$) into a misfolded and infectious form termed $PrP^{Sc}$ (7–9). $PrP^{Sc}$ aggregates into oligomers and amyloids that accumulate predominantly in the central nervous system. $PrP^{Sc}$ assemblies usually exhibit increased resistance to protease digestion compared to $PrP^C$. The protease-resistant core of the protein ($PrP^{res}$) serves as a key biochemical marker of the disease. Distinct prion strains, each defined by a characteristic clinic-pathological phenotype, disease duration, and $PrP^{res}$ electrophoretic profile, are identified in specific hosts (10). These strain-specific traits are encoded by structurally distinct conformers of $PrP^{Sc}$. Human sCJD displays a broad clinico-pathological diversity, driven by the combination of multiple prion strains and host polymorphism at codon 129 of the prion protein gene (*PRNP*), which encodes either methionine (M) or valine (V). $PrP^{res}$ in sCJD cases is classified as type 1 (displaying a 21 kDa unglycosylated fragment) or type 2 (displaying a 19 kDa unglycosylated fragment) (11, 12). Based on the $PrP^{res}$ type, codon 129 genotype, and histopathology, sCJD is categorized into at least seven molecular subtypes: MM1, MV1, VV1, MM2 (thalamic and cortical forms), MV2, and VV2. Experimental transmission studies using humanized transgenic mice expressing human PrP have shown that these subtypes are (at least partially) associated with distinct prion strains (13). Based on the classification by Bishop et al., which considers codon 129 genotype and $PrP^{res}$ electrophoretic signature in the donor brain, M1 and V2 strains account for most MM1/MV1 and VV2/MV2 cases, respectively, while the V1 strain corresponds to VV1, and M2-c and M2-t correspond to MM2-cortical and MM2-thalamic forms. Interestingly, transmission studies in certain humanized transgenic mouse lines have shown that VV2 and MV2 subtypes produced distinct biological phenotypes, suggesting that different prion strains may underlie these grouped subtypes (14). Moreover, co-propagation of M1 and V2 strains has been reported in numerous MM1/MV1 and MV2/VV2 cases (15), highlighting the complexity of strain diversification and interactions within the brain of sCJD-affected individuals (16).

Clinical diagnosis of CJD primarily relies on the observation of rapidly progressing neurodegenerative symptoms, including dementia, supported by magnetic resonance imaging of the brain (17). However, definitive confirmation still requires post-mortem histopathological and biochemical analyses (18). *In vitro* amplification techniques have recently emerged as powerful tools for prion detection, capable of detecting minute amounts of $PrP^{Sc}$ assemblies with high sensitivity and specificity. Protein misfolding cyclic amplification (PMCA) is one such technique; it uses $PrP^C$ derived from uninfected brain homogenates (BH; usually from transgenic mice expressing various mammalian $PrP^C$ isoforms) as a substrate (19). While PMCA effectively amplifies vCJD prions, it shows limited efficacy with certain prions such as MM1 sCJD (20, 21). A second technique, real-time quaking-induced conversion (RT-QuIC), employs recombinant PrP expressed in *Escherichia coli* as a substrate (22–24). RT-QuIC amplifies all sCJD strains but has greater difficulty amplifying vCJD (25, 26).

RT-QuIC offers several advantages over PMCA. It enables real-time detection through the fluorescence of Thioflavin T, whose properties change upon binding to amyloid structures. Prions from BH can typically be detected within 48 hours. Unlike PMCA, RT-QuIC requires fewer steps, avoiding the need for repeated amplification cycles, proteinase K digestion, and western blot analysis. Its simplicity makes it highly suitable for diagnostic applications and anti-prion drug screening (27). Furthermore, RT-QuIC mixtures appear non-infectious (28), reducing biosafety concerns compared to PMCA. RT-QuIC is also highly versatile, supporting detection from cerebrospinal fluid (CSF), blood, nasal brushes, saliva, urine, feces, skin, muscles, and tears (23, 29–36). RT-QuIC

using CSF has even been incorporated into the diagnostic criteria for sCJD at several surveillance centers (17, 37–39). Nonetheless, RT-QuIC has limitations. Discriminating between vCJD and MM1 sCJD typically requires multiple amplification assays (40), and overall RT-QuIC does not yet reliably differentiate all human prion strains, a significant drawback given the influence of strains on disease progression. Strain variability also impacts the efficacy of anti-prion therapies, which are often strain-specific (41). Early prion detection is therefore critical for both patient management and limiting secondary transmission risks, for instance, after invasive surgery. Rapid and reliable prion strain typing is also essential for prompt enrollment and stratification of patients in clinical trials. Thus, a new RT-QuIC substrate should ideally combine robust detection across all human prion strains, including amplification-resistant subtypes, with the ability to discriminate between strains. Achieving this goal would significantly advance prion diagnostics and patient care.

Here, we present the E219K polymorphism of the human PrP M129 allele (PrP E219K) as a promising RT-QuIC substrate. This dominant-negative polymorphism, known to confer protection against sCJD in Asian populations (42), demonstrates significant potential for improving prion detection and discrimination. PrP E219K reliably amplifies prions from both humanized transgenic tg650 mouse BH and directly from patient samples, initiating amplification between 3 and 36 hours for tg650-passaged sCJD, 11 and 31 hours for tg650-passaged vCJD, 0 and 27 hours for patient-derived sCJD, 17 and 35 hours for patient-derived vCJD, over a 5- to 8-log dilution range depending on the strain. VV1 patient-derived prions were amplified over only a 2-log range. Importantly, PrP E219K enabled clear discrimination of vCJD and all tested sCJD strains based on lag time analysis of the amplification reactions. It also discriminated MM2-c sCJD from other sCJD strains and MM1 from MV1 strain despite their phenotypic similarities in humanized mice. These findings demonstrate the versatility and potential clinical utility of PrP E219K in addressing key challenges in prion strain detection and patient management.

## MATERIALS AND METHODS

### Brain homogenates and western-blot analysis of PrP^Sc

BHs were prepared in 5% (wt/vol) glucose from tg650 mice infected with sCJD or vCJD prions (as pooled BHs from multiple animals) and from individual sCJD and vCJD patient samples. PrP$^{res}$ was extracted from 20% BHs (mouse) or 10% BHs (human patients) using the Bio-Rad TeSeE detection kit, as previously described (43, 44). Briefly, 200 µL aliquots were digested with proteinase K (final concentration 200 µg/mL final concentration in buffer A) for 10 min at 37°C, followed by precipitation with buffer B and centrifugation at 28,000 × $g$ for 5 min. Pellets, along with a serial dilution of purified recombinant human PrP, were mixed with Laemmli sample buffer, denatured, and separated on 12% Bis-Tris Criterion gels (Bio-Rad, Marne la Vallée, France) and electrotransferred onto nitrocellulose membranes. The membranes were probed with the biotinylated anti-PrP monoclonal antibody Sha31 (targeting residues 145–152 of human PrP; 0.1 µg/mL) (45), followed by horseradish peroxidase-conjugated streptavidin. Immunoreactivity was visualized by chemiluminescence (Pierce ECL, Thermo Scientific, Montigny le Bretonneux, France; Fig. S1). PrP$^{res}$ size and abundance were determined by quantifying all three glycoforms using Image Lab software and a calibration curve generated from purified recombinant human PrP. Chemiluminescent signals were acquired with the Chemidoc digital imager (Bio-Rad, Marne la Vallée, France). The amount of brain loaded per lane was expressed as "brain equivalent" (mg), i.e., the calculated mass of the original processed tissue in the sample volume.

### Real-time quaking-induced conversion

RT-QuIC amplifications were performed as previously described (46, 47). In brief, 1 µL of 10% human BHs or 20% mouse BHs was subjected to serial 10-fold dilutions in

PBS-SDS-N2 buffer (20 mM sodium phosphate buffer pH 7.4, 130 mM NaCl, 0.1% SDS, and 1× N2 supplement [Thermo Fisher, France]). Then, 2 µL of each dilution was added to 98 µL of reaction buffer (20 mM sodium phosphate buffer, pH 7.4, 300 mM NaCl, 10 µM thioflavin T, 1 mM EDTA, and 100 µg/mL purified recombinant E219K human PrP, $M_{129}$ allele) in a black 96-well optical-bottom plate (48). Plates were sealed with Nunc Amplification Tape (Nalgene Nunc International, France), placed in a Xenius XM spectrofluorometer (Safas, Monaco), and incubated at 46°C ± 2°C for 24–60 hours. Reactions alternated between 1 min orbital shaking (600 rpm) and 1 min rest cycles, with fluorescence measurements recorded every 30 min. All reactions were performed in quadruplicate, except for tg650-negative controls, which were run in octuplicate. Each amplification curve was fitted using MATLAB (R2022b, MathWorks) with the following equation:

$$I = Imin + \frac{(Imax - Imin)X^h}{\left(K + X^h\right)},$$

where $I$ is the fluorescence intensity, and $X$ is time. The parameters derived from this fit included the increase in fluorescence intensity, the slope at the inflection point, and the lag time (defined by the intersection of the tangent at the inflection point with the baseline Imin) (46, 47). The seeding activity titer ($SD_{50}$; the seeding dose yielding thioflavin-T positivity in 50% of replicates) was calculated using the Spearman-Kärber method (49). If less than 100% of RT-QuIC reactions seeded with the first dilution were positive or if no dilution yielded 100% positivity, a trimmed variant of the Spearman-Kärber method was applied (50). The titer was expressed as $SD_{50}$ per mg of the brain. For comparison of strains or patient isolates, the area under the curve (AUC) of lag time (of RT-QuIC amplification reactions) vs particle concentration was computed for each sample. Statistical significance was assessed using a linear model with a fixed strain effect applying both classical and permutation tests (with lmPerm package [51] in R) to assess the robustness of the results given the limited number of replicates. $P$-values were adjusted for multiple comparisons using the Tukey method.

## RESULTS

### PrP E219K RT-QuIC amplifies tg650-passaged sporadic CJD prions

We first evaluated PrP E219K as a substrate for RT-QuIC amplification experiments using BHs from terminally ill humanized tg650 mice (expressing the PrP M129 allele). These mice were serially inoculated (4–5 passages) with MM1, MV1, MM2-c, MV2, VV2, or VV1 sCJD patient isolates (43, 52). For clarity, the resulting tg650-passaged prions are referred to as tg650-MM1, tg650-MV1, tg650-VV1, tg650-MM2-c, tg650-MV2, and tg650-VV2. In tg650 mice, MM1 and MV1 cases produced similar phenotypes, consistent

**TABLE 1** RT-QuIC metrics, PrP$^{res}$ levels, and infectivity titers for human prions passaged in tg650 mice[a]

| Prion strains | tg650-MM1 | tg650-MV1 | tg650-VV1 | tg650-MM2-c | tg650-MV2 | tg650-VV2 | tg650-vCJD |
|---|---|---|---|---|---|---|---|
| **RT-QuIC metrics** | | | | | | | |
| Lag time range (h) | 3–24 | 3–15 | 3–14 | 5–36 | 4–21 | 3–21 | 11–31 |
| Dilution range | >7 log | 5 log | 7 log | 6 log | 5 log | 6 log | 6 log |
| First positive dilution | $2 \times 10^{-3}$ | $2 \times 10^{-4}$ | $2 \times 10^{-2}$ | $2 \times 10^{-3}$ | $2 \times 10^{-3}$ | $2 \times 10^{-3}$ | $2 \times 10^{-3}$ |
| Last positive dilution | $>2 \times 10^{-9}$ | $2 \times 10^{-8}$ | $2 \times 10^{-8}$ | $2 \times 10^{-8}$ | $2 \times 10^{-7}$ | $2 \times 10^{-8}$ | $2 \times 10^{-8}$ |
| SD$_{50}$ (per mg of brain) | $7.91 \times 10^8$ | $4.45 \times 10^7$ | $4.45 \times 10^7$ | $2.50 \times 10^7$ | $1.47 \times 10^7$ | $7.91 \times 10^7$ | $7.91 \times 10^7$ |
| **PrP$^{res}$ levels** | | | | | | | |
| PrP$^{res}$ (ng/mg of brain) | 18.2 | 7.7 | 32.9 | 4.5 | 45.6 | 36.3 | 860.4 |
| **Infectious titer** | | | | | | | |
| ID$_{50}$ (per mg of brain) | $5.01 \times 10^6$ (46) | nd | nd | $1.26 \times 10^5$ (43) | $2.29 \times 10^5$ | nd | $6.31 \times 10^5$ (44) |

[a]Key RT-QuIC amplification metrics for each prion tested in tg650 mice include lag time ranges, dilution ranges, first and last positive dilutions, and the seeding dose (SD$_{50}$) per mg of brain. PrP$^{res}$ accumulation (ng/mg of brain) was quantified by immunoblot, as detailed in the "Materials and Methods" section. Infectivity titers (ID$_{50}$/mg of brain) were determined by titration by limiting dilution in tg650 mice, where available (43, 44, 46) (nd: not determined).

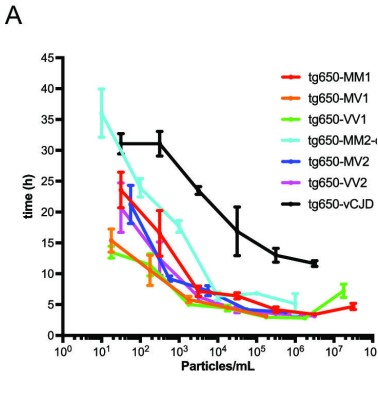

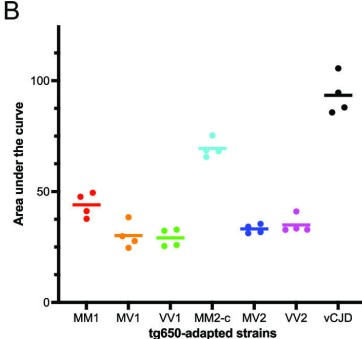

**FIG 1** RT-QuIC analyses of BHs from human PrP tg650 mice infected with sCJD or vCJD prions. (A) Lag times vs particle concentration for amplification reactions seeded with tg650-sCJD BHs and tg650-vCJD BH. (B) AUC of lag time vs particle concentration for each prion strain.

with infection by the same M1 strain (52). The other subtypes yielded distinct phenotypes, indicative of at least five additional strains (43, 52). BHs from uninfected tg650 mice served as negative controls. To test amplification, we prepared 10-fold serial dilutions (spanning eight logs) from 20% BH in seed dilution buffer (PBS-SDS-N2) and initiated RT-QuIC reactions with 2 µL of each dilution. The mean of fitted fluorescence intensity of positive reactions at each dilution for each strain over time is shown in Fig. S2. Lag times were calculated from the fitted curves for each positive replicate. In addition, $SD_{50}$ and the concentration of amplifiable particles were estimated using the Spearman-Kärber equation for each strain (Table 1). Since RT-QuIC prion amplification is highly substrate dependent, comparing amplification reactions at equivalent amplifiable particle concentrations allows normalization across strains and avoids confounding due to strain-specific differences in aggregate size, size distribution, or structure—factors that complicate comparisons based solely on dilution of the initial material. Figure 1A illustrates the relationship between lag time and particle concentration.

No spontaneous auto-polymerization of PrP E219K was observed during the 50-hour amplification reaction period (Fig. S2H). As shown in Fig. 1A and Fig. S2, all six tg650-sCJD prions were efficiently amplified using PrP E219K across broad dilution ranges: from at least five logs for tg650-MV1 and tg650-MV2 to more than seven logs for tg650-MM1 (Table 1). All the strains except tg650-VV1 (Fig. S2C) exhibited inhibition of the amplification reaction at the highest BH concentrations (500-fold dilution of BHs, and even at 5,000-fold dilution for tg650-MV1), a phenomenon previously reported in the literature (53). Analysis of lag time as a function of particle concentration showed that lag times generally decreased as particle concentration increased, except at concentrations exceeding $10^7$ particles per mL, where the aforementioned amplification inhibition occurred. Across all strains (excluding tg650-MM2-c at the limiting dilution), amplification reactions began between 3 and 24 hours, with most reactions starting

**TABLE 2** Statistical comparison of tg650-CJD prion strains[a]

|  | tg650-MV1 | tg650- VV1 | tg650-MM2-c | tg650-MV2 | tg650- VV2 | tg650-vCJD |
|---|---|---|---|---|---|---|
| tg650-MM1 | 0.0195 | 0.0106 | <0.0001 | 0.1022 | 0.2444 | <0.0001 |
| tg650-MV1 | – | 1.0000 | <0.0001 | 0.9826 | 0.8507 | <0.0001 |
| tg650- VV1 | – | – | <0.0001 | 0.9290 | 0.7064 | <0.0001 |
| tg650-MM2-c | – | – | – | <0.0001 | <0.0001 | <0.0001 |
| tg650-MV2 | – | – | – | – | 0.9988 | <0.0001 |
| tg650-VV2 | – | – | – | – | – | <0.0001 |

[a]$P$-values from statistical analyses of the AUC of lag time as a function of particle concentration to compare tg650-passaged CJD prion strains. A linear model with permutation tests was used to evaluate differences between strains. Significant differences ($P < 0.05$) indicate strain discrimination. Cells marked with "–" indicate redundant comparisons.

around 5 hours across a 3- to 4-log dilution range. To compare strains quantitatively, we calculated the AUC of lag time vs particle concentration within the same amplification range spanning a 5-log dilution range (Fig. 1B). Statistical analyses were conducted using a linear model with a strain effect, applying both classical and permutation tests to assess the robustness of the results given the limited number of replicates. Both methods yielded similar results. Table 2 reports $P$-values obtained from the permutation test. It shows that tg650-MM2-c could clearly be discriminated from the other tg650-sCJD prion strains. Furthermore, although tg650-MM1 and tg650-MV1 shared similar phenotypes in mice, they exhibited significantly different seeding kinetics, supporting their differentiation by RT-QuIC.

Collectively, these results demonstrate that PrP E219K enables the amplification of multiple tg650-sCJD prion strains and allows some discriminations based on the lag time of the amplification kinetics.

## PrP E219K RT-QuIC discriminates tg650-variant CJD prions from tg650-sCJD prions

We next investigated whether PrP E219K could amplify tg650-passaged variant CJD (tg650-vCJD). Serial 10-fold dilutions (spanning eight logs) were prepared from 20% BH obtained from terminally ill tg650 mice infected with human vCJD prions (44). The mean of fitted fluorescence intensity of positive reactions at each dilution over time is shown in Fig. S2G. As shown in Fig. S2G and Fig. 1A, tg650-vCJD prions were readily amplified using PrP E219K as a substrate, with a dilution range extending over six logs (Table 1). Amplification reactions started between 11 and 31 hours, and the lag times were consistently longer than those of tg650-sCJD at comparable seed concentrations. To evaluate the ability of PrP E219K to distinguish between strains, we applied the same statistical analyses as described earlier to the AUC of lag time vs particle concentration (Fig. 1B, Table 2). These analyses confirm that, based on the amplification kinetics, PrP E219K effectively discriminates tg650-vCJD prions from tested tg650-sCJD prions.

**TABLE 3** RT-QuIC metrics and PrP[res] levels for CJD prions from human patient BHs[a]

| Strain | MM1 | MV1 | VV1 | MM2-c | MV2 | VV2 | vCJD |
|---|---|---|---|---|---|---|---|
| RT-QuIC metrics |  |  |  |  |  |  |  |
| Lag time range (h) | 0–16 | 2–27 | 11–22 | 2–13 | 3–10 | 5–24 | 17–35 |
| Dilution range | 6 log | 8 log | 2 log | 6 log | 6 log | 5 log | 5 log |
| First positive dilution | $10^{-3}$ | $10^{-3}$ | $10^{-3}$ | $10^{-3}$ | $10^{-3}$ | $10^{-3}$ | $10^{-3}$ |
| Last positive dilution | $10^{-8}$ | $10^{-10}$ | $10^{-4}$ | $10^{-8}$ | $10^{-8}$ | $10^{-7}$ | $10^{-7}$ |
| SD$_{50}$ (per mg of brain) | $1.58 \times 10^{10}$ | $8.89 \times 10^{10}$ | $2.81 \times 10^6$ | $2.81 \times 10^{10}$ | $8.89 \times 10^8$ | $5.00 \times 10^9$ | $2.81 \times 10^9$ |
| PrP[res] levels |  |  |  |  |  |  |  |
| PrPres (ng/mg of brain) | 88.4 | 44.0 | nd | 22.2 | 87.8 | 12.2 | 142.7 |

[a]RT-QuIC amplification metrics include lag time, dilution ranges, first and last positive dilutions, and seeding dose (SD$_{50}$) per mg of brain. PrP[res] accumulation (ng/mg of brain) was quantified by immunoblot, as detailed in the "Materials and Methods" section. (nd: not detected).

## PrP E219K RT-QuIC amplifies sCJD and vCJD prions directly from patient brain homogenates

We next evaluated whether PrP E219K could amplify human prions directly from BHs of patients with sCJD and vCJD. For this analysis, we used the same human BHs that had previously been used to inoculate tg650 mice. An uninfected human post-mortem BH was included as a negative control. Tenfold serial dilutions (spanning nine logs) were prepared from 10% BHs and used to seed the RT-QuIC reactions. The mean of fitted fluorescence intensity of positive reactions at each dilution over time is shown in Fig. S3. No spontaneous auto-polymerization of PrP E219K was observed during the 40-hour amplification reaction period in reactions seeded with the uninfected control BH (Fig. S3H). In contrast, all CJD patient samples generated positive amplification signals when PrP E219K was used as a substrate (Fig. S3 and Table 3). Except for the VV1 sample, all sCJD patient BHs yielded positive reactions over a wide dilution range (≥5 logs). The vCJD patient sample similarly was amplified across a 5-log dilution range. Lag times were calculated from the fitted fluorescence curves for each positive reaction. Using the Spearman-Kärber method, we estimated the $SD_{50}$ values and calculated the amplifiable particle concentrations for each patient sample (Table 3). Figure 2A illustrates lag time as a function of particle concentration.

Consistent with observations from tg650-passaged prions, the vCJD patient sample exhibited significantly longer lag times than sCJD samples at comparable seed concentrations (Fig. 2A). To further compare these strain-specific kinetics, the AUCs of lag time as a function of particle concentration were calculated under two conditions: a short dilution range including VV1 prions (Fig. 2B) and a broader dilution range excluding VV1 (Fig. 2C). Statistical comparisons were conducted using a linear model with a strain effect, applying both classical and permutation tests to assess the robustness of the results given the limited number of replicates. Both methods yielded similar results. Table 4 reports the P-values obtained from the permutation test, confirming that vCJD can be clearly distinguished from tested sCJD based on the lag time of the amplification kinetics.

These findings demonstrate that PrP E219K is a robust and versatile substrate for RT-QuIC amplification of both sporadic and variant CJD prions. Importantly, the lag time parameter provides a reliable and discriminating metric to distinguish between tested sCJD and vCJD prion isolates.

## DISCUSSION

RT-QuIC is emerging as one of the most relevant assays for the pre-mortem diagnosis of sporadic CJD, with applications ranging from disease surveillance and differential diagnosis to the identification of at-risk tissues and the enrollment of patients into therapeutic trials during prodromal phases. However, the heterogeneous phenotypic spectrum of sCJD, shaped by host genetic factors and the existence of multiple prion strains, challenges RT-QuIC diagnostic performance, particularly for rarer sCJD subtypes. Moreover, while RT-QuIC has proven valuable in detecting sCJD, its ability to differentiate sCJD from vCJD remains limited. Amplification of vCJD prions often shows reduced sensitivity and may require multiple substrates to distinguish them from the MM1 sCJD prions (40). Although clinical vCJD cases remain rare, the potential for a large number of asymptomatic carriers in whom peripheral tissues harbor PrP[Sc] (54) underscores the urgent need for reliable diagnostic markers, especially for use with peripheral tissues.

The major recombinant PrPs currently used as RT-QuIC substrates include hamster PrP, human PrP, chimeric hamster-human PrP, truncated hamster PrP, and bank vole PrP. In this study, we demonstrate that recombinant human PrP E219K, expressed in *E. coli*, is a promising alternative substrate for rapidly amplifying and detecting all human sCJD subtypes as well as vCJD prions. PrP E219K addresses some key limitations of current RT-QuIC substrates, particularly their sensitivity to auto-polymerization. In our experiments, reactions seeded with dilutions of BHs from uninfected human or transgenic mouse brains showed no auto-polymerization over ≥40 hours. As a result, we

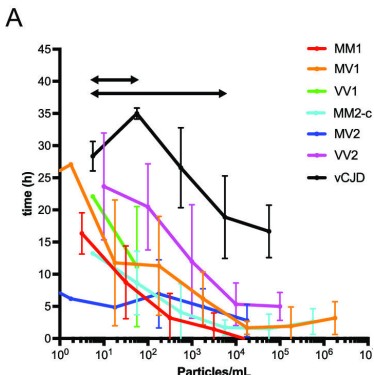

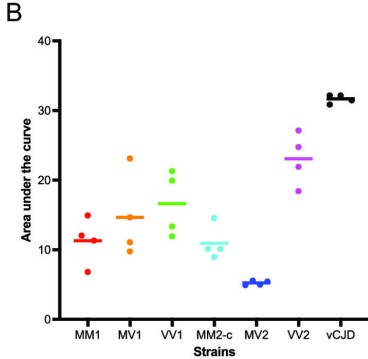

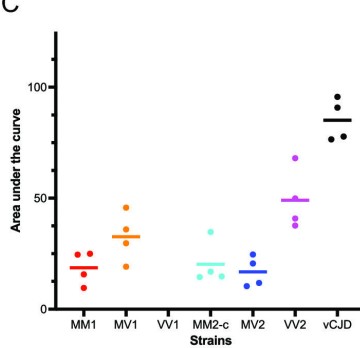

**FIG 2** RT-QuIC analyses of BHs from patients with sCJD or vCJD prions. (A) Lag time vs particle concentration for amplification reactions seeded with sCJD and vCJD patients' BHs. (B) AUC of lag time vs particle concentration for each strain over a 2-log dilution range. (C) AUC of lag time vs particle concentration for each strain over a 4-log dilution range.

observe no false-positive reactions, allowing straightforward and rapid identification of positive samples. This contrasts with other recombinant substrates, for which spontaneous auto-polymerization often complicates interpretation (55–57).

Our results show that PrP E219K successfully amplifies all sCJD subtypes tested (MM2-t was not available) and vCJD prions, using both tg650-passaged materials and direct human brain homogenates. All tested human prions, except the VV1 patient-derived sample, were amplified over a dilution range of at least five logs, with most achieving 6–7 logs. The VV1 patient sample could be amplified over only two logs compared to seven logs for the corresponding tg650-passaged strain. This limited amplification warrants further investigation, particularly concerning sample characteristics such as brain region sampled, tissue preservation, or disease stage. Supporting this, our western blot analyses revealed markedly lower PrP^res levels in the VV1 patient brain, suggesting that reduced seeding material likely contributed to limited amplification.

**TABLE 4** Statistical comparison of CJD prions from patient BH[a]

| | MV1 | VV1 | MM2-c | MV2 | VV2 | vCJD |
|---|---|---|---|---|---|---|
| | 0.8306 | 0.3835 | 1.0000 | 0.2525 | 0.0023 | <0.0001 |
| MM1 | 0.3925 | – | 0.9999 | 0.9998 | 0.0048 | <0.0001 |
| | – | 0.9846 | 0.7628 | 0.0185 | 0.0437 | <0.0001 |
| | – | – | 0.5139 | 0.2696 | 0.2330 | <0.0001 |
| MV1 | – | – | 0.3167 | 0.0032 | 0.1963 | 0.0001 |
| VV1 | – | – | – | – | – | – |
| | – | – | – | 0.3106 | 0.0017 | <0.0001 |
| MM2-c | – | – | – | 0.9964 | 0.0077 | <0.0001 |
| | – | – | – | – | <0.0001 | <0.0001 |
| MV2 | – | – | – | – | 0.0028 | <0.0001 |
| | – | – | – | – | – | 0.0368 |
| VV2 | – | – | – | – | – | 0.0009 |

[a].P-values from statistical analyses of the AUC of lag time as a function of particle concentration to compare human BHs infected with sCJD and vCJD prions. A linear model with permutation tests was used to evaluate differences between human prions. The first row for each pairwise comparison corresponds to analyses of the AUC over a 2-log dilution range; the second row corresponds to analyses of the AUC over a 4-log dilution range. Significant differences ($P < 0.05$) indicate strain discrimination. Cells marked with "–" indicate redundant comparisons.

This reduced sensitivity is consistent with previous reports highlighting the difficulty in detecting VV1 and MM2 prions by RT-QuIC (58). Expanding the sample cohort and optimizing RT-QuIC conditions may improve sensitivity for such underrepresented subtypes. Unfortunately, access to VV1 samples remains extremely limited due to the rarity of this subtype.

Using endpoint dilution and PrP$^{res}$ quantification, we estimated the lower detection limit of PrP E219K in our RT-QuIC assay. Based on the last positive dilution, PrP$^{res}$ detection thresholds reached ~1 fg for human MM1, tg650-VV1, human and tg650-VV2, and human and tg650-MV2, 0.2 fg for tg650-MV1 and human and tg650-MM2-c, <0.05 fg for tg650-MM1 and human MV1, and approximately 30 fg for human and tg650-vCJD prions per reaction. Of note, the detection limits are similar for human and tg650-passaged CDJ prions, except for MM1 and MV1 prions, for which tg650-MM1 and human MV1 displayed a lower detection limit. These limits were similar to or lower than those reported for other RT-QuIC substrates (typically ~1 fg for sCJD MM1, MM2, and MV2 [23, 26, 53]). Notably, vCJD prions were detected using PrP E219K at amounts five times lower than previously reported using other substrates (26). These performance levels were achieved without sample enrichment or secondary amplification steps, which can increase variability and delay the results (25).

Lag time emerged as a powerful parameter for prion strain discrimination. Amplification of sCJD prions (from either patients or tg650-passaged material) typically began between 0 and 7 hours at high particle concentrations (>$10^3$ particles/mL), faster than the 7–20 hours reported for hamster or human PrP substrates (23, 25, 26). Amplifications of vCJD prions began later (between 17 and 35 hours with patient samples and between 11 and 31 hours with tg650-passaged materials), yet lag times were still shorter than the >30 hours often needed when using hamster PrP (26). Shorter lag times offer two key advantages: (i) faster diagnostic results and (ii) reduced risk of auto-polymerization, improving assay sensitivity and data interpretability. Moreover, PrP E219K enables clear differentiation of vCJD from all tested sCJD strains based solely on lag times of the kinetics, without the need for multiple substrates or protease-resistant amplified profile comparisons (40). It also distinguished tg650-MM2-c from other tg650-sCJD strains and successfully differentiated tg650-MM1 from tg650-MV1, despite their similar phenotypic in tg650 mice. This highlights the utility of PrP E219K for prion strain discrimination and the value of kinetic parameters, as previously demonstrated in other models (e.g., tg650-MM1 vs tg650-adapted atypical scrapie [47]).

Our study underscores the potential of PrP E219K to improve prion diagnostics. Expanding the number of clinical samples and optimizing assay conditions will further

refine its performance, especially for rarer or low-seeding strains. In parallel, evaluating its use with alternative biological matrices (e.g., cerebrospinal fluids, blood) could broaden its clinical applicability.

In conclusion, PrP E219K is a robust RT-QuIC substrate capable of rapidly amplifying and detecting a wide range of human prion strains. Its ability to differentiate sCJD from vCJD based on the lag time of the amplification kinetics represents a significant advance for both diagnostic and research applications, including strain discrimination.

## ACKNOWLEDGMENTS

D.M. was supported by a Young Scientist Grant from the Agence Nationale de Sécurité du Médicament et des produits de Santé (grant number 2014 S033 HAP ANSM 2014/iPDB).

A.M.M. was supported by a postdoctoral fellowship from the Fundación Ramón Areces (XXXIV Convocatoria para Ampliación de Estudios en el Extranjero en Ciencias de la Vida y de la Materia). This work was also supported by Agence Nationale de Recherche (ANR-21-CE15-0011-01).

A.M.M., V.B., and D.M. designed the research. A.M.M., F.R., F.J., L.H., H.R., S.H., I.Q., V.B., and D.M. performed the research and analyzed data. A.M.M, V.B., and D.M. wrote the paper. All authors approved the final version of the manuscript.

## AUTHOR AFFILIATIONS

[1]Université Paris-Saclay, Institut National de Recherche pour l'Agriculture, l'Alimentation et l'Environnement, Université Versailles-Saint Quentin, Unité de Virologie et d'Immunologie Moléculaires, Jouy-en-Josas, France
[2]Université Paris-Saclay, Institut National de Recherche pour l'Agriculture, l'Alimentation et l'Environnement, AgroParisTech, Unité de Génétique Animale et Biologie Intégrative, Jouy-en-Josas, France
[3]Biochemistry and Molecular Biology Department, Neurodegenerative Pathologies, LBMMS, Hospices Civils de Lyon, Hôpital neurologique, Lyon, France
[4]BIORAN Team, Lyon Neurosciences Research Center, Lyon, France
[5]Sorbonne Université, Institut du Cerveau - Paris Brain Institute - ICM, Inserm, CNRS, Paris, France

## PRESENT ADDRESS

A. Marín-Moreno, Laboratorio Central de Veterinaria, Ministerio de Agricultura, Pesca y Alimentación, Algete (Madrid), Spain

## AUTHOR ORCIDs

V. Béringue ⓘ http://orcid.org/0000-0001-6706-5712
D. Martin ⓘ http://orcid.org/0000-0003-2421-716X

## FUNDING

| Funder | Grant(s) | Author(s) |
| --- | --- | --- |
| Agence Nationale de Sécurité du Médicament et des Produits de Santé | 2014S033 HAP ANSM 2014/iPDB | D. Martin |
| Fundación Ramón Areces | XXXIV convocatoria para ampliación de estudios en el extranjero de la vida y de la materia | A. Marín-Moreno |
| Agence Nationale de la Recherche | ANR-21-CE15-0011-01 | H. Rezaei |
| | | V. Béringue |
| | | D. Martin |

## AUTHOR CONTRIBUTIONS

A. Marín-Moreno, Conceptualization, Formal analysis, Funding acquisition, Investigation, Validation, Writing – original draft, Writing – review and editing | F. Reine, Investigation | L. Herzog, Investigation | H. Rezaei, Formal analysis, Investigation | V. Béringue, Conceptualization, Formal analysis, Funding acquisition, Investigation, Methodology, Supervision, Validation, Writing – original draft, Writing – review and editing | D. Martin, Conceptualization, Data curation, Formal analysis, Funding acquisition, Investigation, Methodology, Project administration, Resources, Software, Supervision, Validation, Visualization, Writing – original draft, Writing – review and editing.

## DATA AVAILABILITY

All the data generated or analyzed during this study are included in this published article and its supplementary information files.

## ETHICS APPROVAL

MM1-sCJD (UK NHBX0/0001), MV2-sCJD (UK NHBX0/0004) and vCJD (NHBY0/0003) samples were provided by the UK National Institute for Biological Standards and Control (NIBSC) CJD Resource Centre. MV1-sCJD Fr3 (A990055), VV2-sCJD Fr2 (A001002), VV1-sCJD Fr1 (367) and MM2-c-sCJD Fr1 (447) and uninfected brain sample (99-298) were provided by our collaborators (S.H. and I.Q.) as part of the French National Center of Reference for Unconventional Transmissible Agents and the French National Neuropathology Network for CJD, based on the availability of autopsy-retained frozen brain material. For each case, informed consent was obtained from the patient's relatives for genetic analysis of the PrP gene (PRNP), including codon 129 genotyping and exclusion of pathogenic mutations. The patient's relatives also provided written informed consent for autopsy and use of post-mortem tissues in research, in compliance with French regulations (L.1232-1 to L.1232-3, Code de la Santé Publique).

## ADDITIONAL FILES

The following material is available online.

### Supplemental Material

**Data Set S1 (Spectrum00292-25-s0001.xlsx).** Raw data of the RT-QuIC amplification reactions.
**Supplemental figures (Spectrum00292-25-s0002.pdf).** Figures S1 to S3.

### Open Peer Review

**PEER REVIEW HISTORY (review-history.pdf).** An accounting of the reviewer comments and feedback.

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
