## [Reviewer comments · Microbiology Spectrum]

Microbiology Spectrum

Human PrP E219K: a new and promising substrate for robust RT-QuIC amplification of human prions with potential for strain discrimination

Alba Marín-Moreno, Fabienne Reine, Florence Jaffrézic, Laetitia Herzog, Human Rezaei, Isabelle Quadrio, Stéphane Haïk, Vincent Béringue, and Davy Martin

Corresponding Author(s): Davy Martin, Université Paris-Saclay, Institut National de Recherche pour l'Agriculture, l'Alimentation et l'Environnement, Université Versailles-Saint Quentin, Unité de Virologie et d'Immunologie Moléculaires

Review Timeline:

Submission Date:	January 29, 2025
Editorial Decision:	February 24, 2025
Revision Received:	May 12, 2025
Accepted:	June 6, 2025

Editor: Isaac Solomon

Reviewer(s): The reviewers have opted to remain anonymous.

Transaction Report:

DOI: <https://doi.org/10.1128/spectrum.00292-25>

Re: Spectrum00292-25 (Human PrP E219K: a new and promising substrate for robust RT-QuIC amplification of human prions with potential for strain discrimination)

Dear Dr. Davy Martin:

Thank you for the privilege of reviewing your work. Below you will find instructions from the Spectrum editorial office and the reviewer comments.

Revision Guidelines

Sincerely,
Isaac Solomon
Editor
Microbiology Spectrum

Reviewer #1 (Comments for the Author):

This study examines a polymorphism, PRNP E219K, as a novel substrate for RT-QuIC amplification for CJD classification with the potential for strain discrimination. Brain homogenates from patients diagnosed with several different strains of CJD were used as inocula for transgenic mice, followed by 10-fold serial dilutions across 8 logs. Fluorescence intensity was measured over time, with strain discrimination inferred from lag time. Notable discrimination was determined in vCJD strains compared to sCJD strains. Fluorescence was also measured directly on patient brain homogenates with replicated discrimination between vCJD and sCJD strains. The robust discrimination between vCJD and sCJD prion strains demonstrated here, as well as the

improved amplification time (compared to 48 hours with other substrates), is a significant improvement from current postmortem CJD RT-QuIC diagnostic approaches.

Major Concerns

- 1) Page 3, Line 13: The phrases "within a few hours" and "a broad dilution range" are misleading, as strains varied from a lag time of 5-31 hours in mice to <10-25 hours in patient samples. In particular, the vCJD strains had a notably longer lag time in both humanized mice and patient brain homogenates. Moreover, while many strains amplified over several dilution factors, the VV1 strain notably only amplified over a 2-log dilution range. A more precise description of the findings (perhaps separating humanized mice from patient sample findings) is needed.
- 2) Page 7, Line 15: Given the VV1 findings, "2 to 8 logs" would be a more accurate description and "within 20 hours" is not reflective of other language used throughout. For example, "amplification reactions began within 25 hours" (page 11, line 14), "began 11 and 31 hours" (page 12, line 13), and "approximately 25 hours" (page 13, line 20) are used later in the manuscript. Making this language more reflective of the findings is essential.
- 3) Page 10, Line 1: How many mice are being examined per strain? It would be worth mentioning either here or in the methods.
- 4) Figure 1: In part A, particles/mL should start with the lowest dilution (greatest number of particles/mL) on the left and the highest dilution on the right, especially considering the wording on page 11, line 12 ("generally increased" would seem more intuitive to the reader if the graph had a positive slope). In Figure 2, vCJD is combined with the rest of the strains, so it is a bit redundant to separate vCJD from the rest of the strains here in Figure 1. Consider combining part C with the same graph as part A, or moving part B to part C. It is a bit confusing having the AUC analysis for vCJD before vCJD is mentioned. This may require the wording of the experimental process to differ in the text.
- 5) Table 1: It appears that the first positive dilution for tg650-MV1 is 2.10⁻⁴ and the last positive dilution is 2.10⁻⁸ (Supplementary Figure 1C). This should be clarified. In some instances, the SD50, PrPres, and ID50 do not quite align, as in tg650-MM1 (which possesses the highest SD50, yet has comparatively lower PrPres levels), and tg650-vCJD (which has the highest PrPres levels, but possesses the same SD50 as mice with ~20x less PrPres).
- 6) Table 2: While there is clear discrimination between vCJD and sCJD strains, the lack of discrimination between MV2 and VV2 strains from most other strains warrants further discussion.
- 7) Include lag times in Tables 1 and 3 for the first and last positive dilution of each strain.
- 8) Page 15, Line 24: While it appears that using PrP E219K shows promise as a substrate for RT-QuIC, whether it effectively addresses the criteria mentioned earlier (page 7, line 5) is debatable. The amplification lag times vary substantially between strains, while some dilution ranges for each strain are more robust than others. MV2 and VV2 had less strain discrimination capabilities than other strains in mice, while the MM1 and MM2-c strains had less discrimination capabilities in human samples. Notably, there was worse performance of MM2-c, MM1, and MV1 strain discrimination in patient brain samples than in transgenic humanized mice, which is partially acknowledged on page 7, line 18. The VV1 patient sample should be replicated with additional VV1 samples to better understand the reduced sensitivity demonstrated here.
- 9) Page 16, Line 2: The human BH from uninfected controls were amplified over 40 hours, not 50. Change to {greater than or equal to} 40 hours.
- 10) Page 16, Line 12: It would be beneficial to add an additional figure (main or supplementary) containing clinicopathologic data for the patient brains used in this study, particularly since the same brains were used as inocula for the transgenic mice experiments. Useful data would include patient age, brain region examined, sex, postmortem interval, initial symptoms, disease duration, pathology severity, spongiosis and gliosis findings, as well as representative photomicrographs with hematoxylin and eosin staining.
- 11) An additional figure comparing the RT-QuIC amplification reactions used to diagnose the patient samples used in this study with the RT-QuIC used here with E219K substrate could be an interesting comparison, especially, if the E219K substrate is positioned as a faster and more robust substrate for strain discrimination than current RT-QuIC substrates.
- 12) Page 17, Line 21: The PrP E219K substrate was not used in MM2-t strains, so using "all human CJD prion strains" is not entirely accurate. Moreover "rapidly amplifying" is misleading considering the wide variability between vCJD and sCJD.

Minor Concerns

- 1) Page 5, Line 24: Remove "of" between "one" and "such."
- 2) Page 6, Line 1: Consider removing "of" or adding "amounts" after "various" to improve readability.
- 3) Page 6, Line 2: Consider changing "amplify" to "amplifies."
- 4) Page 6, Line 21: Consider changing "required" to "requires."
- 5) Page 9, Line 24: There is no mention of MM2-t humanized mice. Considering that this strain of prion disease is mentioned earlier in the introduction (page 5), a MM2-t humanized mice strain or patient sample should be considered or an explanation given as to why it was not examined in this study.
- 6) Page 11, Line 9: Consider citing Supplementary Figure 1E here for clarity.
- 7) Page 11, Line 16: Specify the amplification range used for AUC measurements.
- 8) Page 11, Line 24: Replace "a" after "degree" with "of."
- 9) Page 12, Line 16: Correct typo to "lag time."
- 10) Supplementary Figure 1: Mention each strain in the order that it appears in Figure 1 and in table 1 for cohesivity. Consider moving this figure to Supplementary Figure 2, since the western blot findings are mentioned first in the methods.
- 11) Supplementary Figure 2: Since this is mentioned first in the methods, I would recommend moving to Supplementary Figure 1. Consider numbering the lanes in the gel to improve readability. Perhaps mention why different mg of brain equivalents were loaded for different strains and how this relates to the variation observed in Table 1, particularly in PrPres levels.

- 12) Page 13, line 14: Change "> 5 logs" to "{greater than or equal to} 5 logs."
- 13) Table 3: Include PrPres levels to better compare to the humanized mice experiments.
- 14) Figure 2: In part A, what do the vertical dotted lines indicate? Presumably, these are the dilution ranges for AUC calculations in parts B and C. Consider specifying this in the legend. However, the dotted lines make this figure busy and hard to interpret. Consider communicating this in a more simplistic way. In part B, a 2-log dilution range is a very small window for comparison. Ideally, I would recommend repeating this with another VV1 patient sample so that no data are excluded in analysis. In part C, only 4 logs of dilutions are being compared. However, according to Table 3, 5 logs could be compared when VV1 is excluded.
- 15) Page 15, Line 18: Remove "clinical cases" after "vCJD" due to redundancy.
- 16) Page 16, Line 4: Change "substrate" to "substrates."

Reviewer #2 (Comments for the Author):

The paper states that:

- 1) The PrP E219K substrate amplifies all 6 sCJD strains and vCJD strain over a broad dilution range within a few hours
- 2) The discrimination of sCJD subtype from vCJD relies on lag time

I agree with 1) but have reservations about statement 2) related to how these data show that the lag time varies with dilution factor.

Although a measurement of PrPres is described in the methods section, the section goes on to state that the RT-QuIC inocula were prepared as 10% dilutions - so there was no standardisation of seed mass between subtypes in the seeding volume.

Line 1 on page 9 says "all experiments were performed in quintuplicate" however looking at the result figures I think this should say "sample replicates" and not "experiments" as I think the results as presented are too clean to be the average of 5 RT-QuIC experiments. If those figures as presented are indeed the average results of 5 separate RT-QuIC experiments, then I would like to see the raw data.

If these are replicates and not distinct experiments, then the results presented are from 1 dilution series on each subtype. I think the differences in lag times are more likely due to the seed mass in the inocula rather than any intrinsic property of the E219K PrP substrate. I suspect that if the seed mass in the inocula are standardised, the differences in lag time between subtypes will disappear.

Other comments:

I am unclear how the models were used for fitting, how they were fitted or the value in using this approach over another. Although I haven't used nlme or lmPerm, the documentation for these packages reads like multiple models are available within these packages, so saying "we used this package" doesn't add value to the section and introduces more questions than answers.

Table 2 - there is one p value in each box but the legend says 2 models were used to "evaluate differences" (in what?! presumably lag times? or goodness of fit?) so please give more details about this.

There is a mixture of references to 40 hours and 50 hour duration in the manuscript, why the difference in endpoints between experiments/figures?

p15 line 25 - Ok, no autopolymerisation is shown in the data, but many other RT-QuIC publications show stability of other PrP substrates up to 100 hrs. Why is being stable for 40 hrs such an advantage in this situation?

p16 lines 17 -25. These data haven't been shown.

Fig 2. - Regarding the "mg (brain equivalent)" on the legend what does that mean - is that the amount of PrPres in the loaded sample? or scaled up to the total sample? or the total brain? This is unclear as written. Was the intensity of all PrPres bands in each lane used to determine this mass, or just one (which?) band in each lane? More detail as to how this was done would be useful.

Fig 3. - Not all plots have SD overlaid - why? While the plots are colorful, they will be difficult for colorblind people to parse and are difficult to interpret as currently presented. I would suggest replotting with datapoints marked as e.g. squares, triangles, dotted lines, solid lines, etc. As currently presented, it's really hard to tell which dilutions are 10-3 and which are 10-9. I'm not sure whether all dilutions are present for each subtype. If not, why not?

Typographical errors:

There are some errors and missing words.

P10 line 11 I think should read "Fig 1A-H"

p11 line 24 "degree of discrimination"

p12 line 13 "began at 11 and 31 hrs"

p12 line 16 missing letter - "lag"

p15 line 24 "addresses" (present tense) not addressed

Here are our responses to reviewers' comments and concerns.
They are in blue in the document.
Of note, pages and lines refer to the new clean version of the manuscript.

Reviewer #1 (Comments for the Author):

This study examines a polymorphism, PRNP E219K, as a novel substrate for RT-QuIC amplification for CJD classification with the potential for strain discrimination. Brain homogenates from patients diagnosed with several different strains of CJD were used as inocula for transgenic mice, followed by 10-fold serial dilutions across 8 logs. Fluorescence intensity was measured over time, with strain discrimination inferred from lag time. Notable discrimination was determined in vCJD strains compared to sCJD strains. Fluorescence was also measured directly on patient brain homogenates with replicated discrimination between vCJD and sCJD strains. The robust discrimination between vCJD and sCJD prion strains demonstrated here, as well as the improved amplification time (compared to 48 hours with other substrates), is a significant improvement from current postmortem CJD RT-QuIC diagnostic approaches.

Major Concerns

1) Page 3, Line 13: The phrases "within a few hours" and "a broad dilution range" are misleading, as strains varied from a lag time of 5-31 hours in mice to <10-25 hours in patient samples. In particular, the vCJD strains had a notably longer lag time in both humanized mice and patient brain homogenates. Moreover, while many strains amplified over several dilution factors, the VV1 strain notably only amplified over a 2-log dilution range. A more precise description of the findings (perhaps separating humanized mice from patient sample findings) is needed.

We thank the reviewer for highlighting this point. We acknowledge that the original phrasing lacked precision and could be misleading. We have revised the manuscript accordingly to ensure consistent terminology when reporting lag time ranges across all prion strains. Specifically, we now explicitly distinguish between results from human patient brain homogenates and those from prions in humanized transgenic mice, both in the main manuscript and in the Abstract section.

Page 7 lines 11 to 14 now reads: "initiating amplification between 3 and 36 hours for tg650-passaged sCJD, 11 and 31 hours for tg650-passaged vCJD, 0 and 27 hours for patient-derived sCJD, 17 and 35 hours for patient-derived vCJD".

In the Abstract (page 3, lines 13–18), we also revised the summary to reflect these precise ranges: "In tg650-passaged prions, amplification reactions initiated between 3 and 36 hours for sCJD prions and between 11 and 31 hours for vCJD prions, covering a 5- to 7-log dilution range depending on the strain. For patient brain homogenates, amplification reactions started between 0 and 27 hours for sCJDs and between 17 and 35 hours for vCJD, covering a 5- to 8-log dilution range depending on the strain."

2) Page 7, Line 15: Given the VV1 findings, "2 to 8 logs" would be a more accurate description and "within 20 hours" is not reflective of other language used throughout. For example, "amplification reactions began within 25 hours" (page 11, line 14), "began 11 and 31 hours" (page 12, line 13), and "approximately 25 hours" (page 13, line 20) are used later in the manuscript. Making this language more reflective of the findings is essential.

We agree that the previous wording lacked consistency and clarity regarding amplification lag times and dilution ranges. We revised the sentence on page 7 to incorporate all relevant data and avoid overly simplified summary language. The updated sentence now reads (page 7, lines 10–15): "PrP E219K reliably amplifies prions from both humanized transgenic tg650 mouse BH and directly from patient samples, initiating amplification between 3 and 36 hours for tg650-passaged sCJD, 11 and 31 hours for tg650-passaged vCJD, 0 and 27 hours for patient-derived sCJD, 17 and 35 hours for patient-derived vCJD, over a 5- to 8-log dilution range depending on the strain. VV1 patient-derived prions were amplified over only a 2-log range."

We also updated the corresponding sections throughout the manuscript to reflect this refined terminology:

- Page 12, lines 4 to 5: "amplification reactions began between 3 and 24 hours",
- Page 13, line 14: "Amplification reactions started between 11 and 31 hours",
- Page 14 lines 22 to 23: "the vCJD patient sample exhibited significantly longer lag times than sCJD samples at comparable seed concentrations.",
- Page 18, line 22 to page 19 line 3: "Amplification of sCJD prions (from either patients or tg650-passaged material) typically began between 0 and 7 hours at high particle concentrations ($>10^3$ particles/mL), faster than the 7-20 hours reported for hamster or human PrP substrates (23, 25, 26). Amplifications of vCJD prions began later, (between 17 and 35 hours with patient samples and between 11 and 31 hours with tg650-passaged materials) yet lag times were still shorter than the >30 hours often needed when using hamster PrP(26)"

We would like to clarify:

- On the former pages 11 and 12, the stated upper limit of 25 hours specifically referred to tg650-sCJD strain.
- On the former page 12 line 13, we reported the lag time range exclusively for the tg650-vCJD strain (between 11 and 31 hours),
- On the former page 13, line 20, we compared patients BHs. In this comparison, we performed a mean of the lag time over dilution range for each strain to make a rapid comparison. This part has been removed as it was not clear and precise enough.

In line with your recommendations, we have now harmonized the descriptions of lag times across the manuscript. We hope this improves the clarity and consistency of our presentation.

3) Page 10, Line 1: How many mice are being examined per strain? It would be worth mentioning either here or in the methods.

As noted in the manuscript, we did not perform new mouse inoculations for this study. Instead, we used materials from previously published inoculation experiments, as referenced in the legend of Table 1 (43, 44, 46) and in the main text page 10 line 6 (43, 52). In those studies, between 6 and 10 mice were inoculated per case, and prions were stabilized through 4 to 5 serial passages in humanized tg650 mice, allowing consistent biological phenotypes to be obtained, as specified on page 10, line 5 of the current manuscript.

43. Chapuis J, Moudjou M, Reine F, Herzog L, Jaumain E, Chapuis C, Quadrio I, Boulliat J, Perret-Liaudet A, Dron M, Laude H, Rezaei H, Beringue V. 2016. Emergence of two prion subtypes in ovine PrP transgenic mice infected with human MM2-cortical Creutzfeldt-Jakob disease prions. *Acta Neuropathol Commun* 4:10.

44. Beringue V, Le Dur A, Tixador P, Reine F, Lepourry L, Perret-Liaudet A, Haik S, Vilotte JL, Fontes M, Laude H. 2008. Prominent and persistent extraneural infection in human PrP transgenic mice infected with variant CJD. *PLoS One* 3:e1419.

46. Martin D, Reine F, Herzog L, Igel-Egalon A, Aron N, Michel C, Moudjou M, Fichet G, Quadrio I, Perret-Liaudet A, Andréoletti O, Rezaei H, Beringue V. 2021. Prion potentiation after life-long dormancy in mice devoid of PrP. *Brain Commun* 3:fcab092.

52. Jaumain E, Quadrio I, Herzog L, Reine F, Rezaei H, Andreoletti O, Laude H, Perret-Liaudet A, Haik S, Beringue V. 2016. Absence of Evidence for a Causal Link between Bovine Spongiform Encephalopathy Strain Variant L-BSE and Known Forms of Sporadic Creutzfeldt-Jakob Disease in Human PrP Transgenic Mice. *J Virol* 90:10867–10874.

4) Figure 1: In part A, particles/mL should start with the lowest dilution (greatest number of particles/mL) on the left and the highest dilution on the right, especially considering the wording on page 11, line 12 ("generally increased" would seem more intuitive to the reader if the graph had a positive slope). In Figure 2, vCJD is combined with the rest of the strains, so it is a bit redundant to separate vCJD from the rest of the strains here in Figure 1. Consider combining part C with the same graph as part A, or moving part B to part C. It is a bit confusing having the AUC analysis for vCJD

before vCJD is mentioned. This may require the wording of the experimental process to differ in the text.

Regarding the clarity and structure of Figure 1, in line with your recommendation, we have merged Panels A and C into a single, consolidated Panel A to streamline the presentation and reduce redundancy, particularly in relation to the data shown for vCJD in Figure 2.

We also considered your suggestion to reverse the X axis (i.e., to place the lowest dilution/highest particle concentration on the left). However, after testing this option, we found that it conflicted with conventional plotting practices, where values typically increase from left to right. Reversing the axis is, in our view, more likely to cause confusion than to aid interpretation. Instead, we revised the corresponding text on page 12 (lines 1 to 3) to improve clarity and match the figure orientation more intuitively: "Analysis of lag time as a function of particle concentration showed that lag times generally decreased as particle concentration increased, except at concentrations exceeding 10^7 particles per mL, where the aforementioned amplification inhibition occurred."

5) Table 1: It appears that the first positive dilution for tg650-MV1 is 2.10^{-4} and the last positive dilution is 2.10^{-8} (Supplementary Figure 1C). This should be clarified. In some instances, the SD₅₀, PrP^{res}, and ID₅₀ do not quite align, as in tg650-MM1 (which possesses the highest SD₅₀, yet has comparatively lower PrP^{res} levels), and tg650-vCJD (which has the highest PrP^{res} levels, but possesses the same SD₅₀ as mice with ~20x less PrP^{res}).

We thank the reviewer for pointing this. You are correct that the first positive dilution for tg650-MV1 is 2.10^{-4} , and not 2.10^{-3} as previously stated and 2.10^{-8} the last positive dilution and not 2.10^{-7} as previously stated. It was an error in Table 1 that has been corrected.

Amplification inhibition at high BH concentrations is a known phenomenon, which has been reported in the literature (e.g., reference (53)). Such inhibition can result from matrix effects, including the presence of residual whole blood, plasma or blood-contaminated tissue components, which are known to interfere with amplification efficiency. To reflect this observation, we have revised the relevant sentence in the Results section (page 11 line 12 to page 12 line 1) to read: "All the strains except tg650-VV1 (Supplementary Figure 2C) exhibited inhibition of the amplification reaction at the highest BH concentrations (500-fold dilution of BHs, and even at 5,000-fold dilution for tg650-MV1), a phenomenon previously reported in the literature(53)."

Regarding the reviewer's second point about inconsistencies between SD₅₀, PrP^{res} levels and ID₅₀ values: these parameters do not always correlate, and this is well established in the prion literature. Therefore, apparent discrepancies between these measures are not unexpected. Several key factors contribute to this:

1. PrP^{res} versus PrP^{Sc}: PrP^{res} represents only the protease-resistant fraction of PrP^{Sc}. Some infectious particles may be protease-sensitive and thus are not detected by immunoblotting (e.g. (Gambetti *et al.* Ann. Neurol. 2008)).
2. Aggregate heterogeneity: prion assemblies are polydisperse and differ across strains in terms of size, size distribution, structure and specific biological activity (Tixador *et al.* plos pathog 2010). For example, two strains with similar PrP^{res} content may differ in infectivity if one is composed mainly of large aggregates with fewer accessible templating interfaces, while the other is dominated by small aggregates with greater amplification potential.
3. Substrate dependence in RT-QuIC: the SD₅₀ determined by RT-QuIC depends on the substrate used. For example, MM1 sCJD prions can be amplified using both hamster 23-231 and bank vole substrates, while vCJD prions are amplifiable only with the bank vole substrate and not with hamster PrP (40). These differences in amplification efficiency are substrate dependent and have been exploited to discriminate between vCJD and MM1 sCJD.

Because of these biological and technical factors, PrP^{res} quantity alone cannot reliably reflect the number of amplifiable particles. For this reason, we chose to estimate the concentration of amplifiable particles (SD₅₀) as a substrate-specific and functionally relevant parameter. This concentration is related to the concentration of templating interfaces and also to the potential interactions of these interfaces with the substrate.

To clarify this rationale for readers, we have added the following sentence to the Results section (page 10, lines 16 to 20): "Since RT-QuIC prion amplification is highly substrate-dependent, comparing

amplification reactions at equivalent amplifiable particle concentrations allows normalization across strains and avoids confounding due to strain specific differences in aggregate size, size distribution or structure, factors that complicate comparisons based solely on dilution of the initial material.”

40. Orru CD, Groveman BR, Raymond LD, Hughson AG, Nonno R, Zou W, Ghetti B, Gambetti P, Caughey B. 2015. Bank Vole Prion Protein As an Apparently Universal Substrate for RT-QuIC-Based Detection and Discrimination of Prion Strains. *PLoS Pathog* 11:e1004983.

6) Table 2: While there is clear discrimination between vCJD and sCJD strains, the lack of discrimination between MV2 and VV2 strains from most other strains warrants further discussion.

As noted in the introduction, RT-QuIC is a powerful and rapid *in vitro* amplification technic, but it does have limitations. A major constraint is its often-limited ability to discriminate between certain prion strains. For example, distinguishing vCJD from MM1 sCJD typically requires multiple amplification runs and analyses (reference 40). This may reflect, in part, the imperfect fidelity of strain amplification *in vitro*, as evidenced by the low or absent infectivity of some RT-QuIC amplification products.

Given these inherent limitations, we chose to focus the discussion on the discriminations that were supported by our statistical analyses, in order to deliver a clear and coherent message. We believe this strategy improves clarity for the reader and aligns with the manuscript main objective. We hope the reviewer will agree with this approach.

It is worth noting that we are currently conducting additional experiments using newly developed substrates designed to enhance strain discriminations. These efforts, however, lie beyond the scope of the present manuscript and will be reported in a future work.

7) Include lag times in Tables 1 and 3 for the first and last positive dilution of each strain.

We thank the reviewer for this suggestion. While we appreciate the interest in including specific lag times for the first and last positive dilutions, we instead chose to report the overall range of lag times observed for each strain in Tables 1 and 3. This decision was made because inhibition effects were frequently observed at the highest concentrations (i.e., lowest dilutions), which complicates interpretation of the “first positive” lag time. As such, reporting a full range provides a more accurate and informative representation of the amplification kinetics across the usable dilution span for each strain. We hope this explanation clarifies our rationale.

8) Page 15, Line 24: While it appears that using PrP E219K shows promise as a substrate for RT-QuIC, whether it effectively addresses the criteria mentioned earlier (page 7, line 5) is debatable. The amplification lag times vary substantially between strains, while some dilution ranges for each strain are more robust than others. MV2 and VV2 had less strain discrimination capabilities than other strains in mice, while the MM1 and MM2-c strains had less discrimination capabilities in human samples. Notably, there was worse performance of MM2-c, MM1, and MV1 strain discrimination in patient brain samples than in transgenic humanized mice, which is partially acknowledged on page 7, line 18. The VV1 patient sample should be replicated with additional VV1 samples to better understand the reduced sensitivity demonstrated here.

We acknowledge that the original sentence (formerly page 15 line 24) may have overstated the capacities of the PrP E219K substrate. We did not intend to imply that this substrate addresses all the challenges associated with RT-QuIC. As such, we have revised the sentence to read: “PrP E219K addresses some key limitations of current RT-QuIC substrates” (page 17, line 12).

As mentioned previously, RT-QuIC allows strain discrimination in selected cases, but not universally across dilutions and strains. In this study, we report that PrP E219K enables discrimination between several prion strains, and we support these findings with statistical analyses, that to our knowledge, have not been performed in previous publications making similar claims.

We agree with the reviewer that amplification lag times can vary considerably across dilutions and strains. That is precisely why, in our comparisons and statistical analyses, we incorporated the full range of observed lag times where possible. This approach allowed us to perform the most robust and representative analysis of strain-specific amplification behaviour.

Regarding the VV1 subtype, we fully agree that additional samples would greatly benefit the study. We have acknowledged this limitation in the Discussion section (page 17 line 23 to page 18 line 7), and also noted that the reduced sensitivity observed for VV1 amplification is consistent with the difficulties reported by other labs (page 18 lines 3 to 5). Importantly, we also analysed PrP^{res} content in the VV1 patient brain homogenate by western blot and found it hardly difficult to detect. This extremely low PrP^{res} level may, at least in part, explain the limited amplification range observed for VV1 in our RT-QuIC assays. Unfortunately, due to the rarity of this subtype, access to additional VV1 remains a challenge.

9) Page 16, Line 2: The human BH from uninfected controls were amplified over 40 hours, not 50. Change to {greater than or equal to} 40 hours.

We thank the reviewer for pointing this out. We have corrected the text to state that amplification of human brain homogenates from uninfected controls lasted ≥ 40 hours, in accordance with the actual experiment duration for these samples.

However, we would like to clarify that all control reactions performed with tg650-passaged prions were run for 50 hours. Importantly, no autopolymerisation was observed during this extended period, which supports the stability of the PrP E219K substrate under our assay conditions. Therefore, although the sentence initially referred to the 50-hour time frame, we believe it was not an overstatement in the context of our full experimental design.

10) Page 16, Line 12: It would be beneficial to add an additional figure (main or supplementary) containing clinicopathologic data for the patient brains used in this study, particularly since the same brains were used as inocula for the transgenic mice experiments. Useful data would include patient age, brain region examined, sex, postmortem interval, initial symptoms, disease duration, pathology severity, spongiosis and gliosis findings, as well as representative photomicrographs with hematoxylin and eosin staining.

The patient brain samples used in this study are well-characterized or reference materials and have been employed in several prior publications. Sample codes are listed in the “Source of human samples, ethics approval and consent to participate” section of the manuscript.

However, because these samples originate from different sources, collected at different times and under varying clinical or institutional practices, the associated clinicopathologic metadata (e.g., brain region examined, postmortem interval, clinical history) are incomplete or inconsistently documented. As such, compiling a comprehensive and standardized dataset—including photomicrographs—for all samples was not feasible.

Given the fragmentary nature of the available data, we felt that including them might not be informative or beneficial to the reader. For transparency, we nevertheless provide the collected metadata to the reviewer, and can add them as a supplemental table upon reviewer request.

patient	sex	age (y)	brain region	postmortem interval (h)	disease duration (months)	First symptoms	Neuropathological pattern
MM1-sCJD (UK NHBX0/0001)	*	*	frontal cortex	*	*	*	*
MV1-sCJD Fr3 (A990055)	F	64	frontal cortex	66	5	Ataxia and visual symptoms (homonymous hemianopia and visual hallucinations)	Severe spongiosis and gliosis in the striatum, the thalamus, the isocortex. Spongiosis in the entorhinal cortex and the molecular layer on the cerebellum. PrP deposits of the synaptic type (3F4 antibody).
VV1-sCJD Fr1 (367)	F	69	cerebellar cortex	24	13	cognitive impairment (language and	*

						behavior disorders)	
MM2-c-sCJD Fr1 (447)	M	62	cerebellar cortex	24	6	cognitive impairment (memory disorder, diplopia) and cerebellar syndrome	*
MV2-sCJD (UK NHBX0:0004)	*	*	frontal cortex	*	*	*	*
VV2-sCJD Fr2 (A001002)	M	54	frontal cortex	*	6	Asthenia and insomnia with early ataxia at first examination	Severe spongiosis and gliosis in the striatum and, to a lesser extent, in the thalamus. The frontal isocortex is slightly involved. Spongiosis and gliosis in the entorhinal cortex and the subiculum. Spongiosis in the molecular layer and rare plaques of the Kuru type in the cerebellum. Synaptic and plaque-like PrP deposits (3F4 antibody) in deep cortical layers and in the molecular and granular layers of the cerebellum.
vCJD (NHBX0/0003)	*	*	frontal cortex	*	*	*	*
healthy (99-298)	*	*	frontal cortex	*	/	/	/

Available clinicopathologic data for the patients. (*: missing data, /: irrelevant)

11) An additional figure comparing the RT-QuIC amplification reactions used to diagnose the patient samples used in this study with the RT-QuIC used here with E219K substrate could be an interesting comparison, especially, if the E219K substrate is positioned as a faster and more robust substrate for strain discrimination than current RT-QuIC substrates.

We agree with reviewer 1 that a comparative analysis between the RT-QuIC reactions used here and those used in clinical diagnostics would be of considerable interest—particularly in evaluating the potential of PrP E219K as a faster and more robust substrate for strain discrimination. Unfortunately, the patient samples used in this study were collected in the early 2000s, long before the implementation of RT-QuIC as a diagnostic tool. As a result, comparative amplification data using other substrates are not available for these cases, and such analyses could not be retrospectively performed.

12) Page 17, Line 21: The PrP E219K substrate was not used in MM2-t strains, so using "all human CJD prion strains" is not entirely accurate. Moreover "rapidly amplifying" is misleading considering the wide variability between vCJD and sCJD.

We agree that the phrasing "all human CJD prion strains" was not entirely accurate, as the MM2-thalamic (MM2-t) subtype was not included in our study. Unfortunately, due to the rarity of MM2-t cases, we did not have access to suitable samples. To correct this, we have either removed the word "all" or added clarifying language throughout the manuscript. Specifically, we made the following revisions:

- Page 3, line 11 "all" removed

- Page 3 line 20 and page 4 line 4: revised to “all tested sCJD subtypes”
- Page 7, line 16: revised to “all tested sCJD strains”
- Page 17, line 20: revised to “all sCJD subtypes tested (MM2-t was not available)”
- Page 19, line 5: revised to “all tested sCJD strain”
- Page 19, line 17: “all” replaced by “a wide range of human prion strains”

Regarding the term “rapidly amplifying,” we acknowledge that amplification lag times vary among strains. However, this term was introduced in the Introduction section (formerly page 6, line 10) to describe RT-QulC in general, in contrast to other prion detection techniques such as PMCA which may take several days to weeks, and bioassays, which require months. Despite variability between strains, the E219K substrate consistently yield results within the 48-hour timeframe typical of RT-QulC. In addition, as discussed on page 18 line 22 to page 19 line 4, some amplification reactions initiated earlier than with previously reported substrates. For these reasons, we retained the term “rapidly” on page 19, line 16, as it accurately reflects the overall timeframe and performance of the assay in the context of the field.

Minor Concerns

1) Page 5, Line 24: Remove "of" between "one" and "such."

Done

2) Page 6, Line 1: Consider removing "of" or adding "amounts" after "various" to improve readability.

Done

3) Page 6, Line 2: Consider changing "amplify" to "amplifies."

Done

4) Page 6, Line 21: Consider changing "required" to "requires."

Done

5) Page 9, Line 24: There is no mention of MM2-t humanized mice. Considering that this strain of prion disease is mentioned earlier in the introduction (page 5), a MM2-t humanized mice strain or patient sample should be considered or an explanation given as to why it was not examined in this study.

This point has been addressed in point 12.

6) Page 11, Line 9: Consider citing Supplementary Figure 1E here for clarity.

Done

7) Page 11, Line 16: Specify the amplification range used for AUC measurements.

Done

8) Page 11, Line 24: Replace "a" after "degree" with "of."

Replaced by “some”

9) Page 12, Line 16: Correct typo to "lag time."

Done

10) Supplementary Figure 1: Mention each strain in the order that it appears in Figure 1 and in table 1 for cohesivity. Consider moving this figure to Supplementary Figure 2, since the western blot findings are mentioned first in the methods.

Done, it has also been performed in Supplementary Figure 3

11) Supplementary Figure 2: Since this is mentioned first in the methods, I would recommend moving to Supplementary Figure 1. Consider numbering the lanes in the gel to improve readability. Perhaps mention why different mg of brain equivalents were loaded for different strains and how this relates to the variation observed in Table 1, particularly in PrPres levels.

As recommended, Supplementary Figure 2 has been moved to become supplementary Figure 1.

Regarding the western-blot, different amounts of brain equivalents were loaded per lane to achieve comparable PrP^{res} signal intensities across the various prion strains. This adjustment was necessary because the amount of PrP^{res} in tg650 brains varies considerably depending on the strain: for example, vCJD-infected tg650 brains contain high levels of PrP^{res}, whereas MM2-c-infected tg650 brains contain lower levels.

The primary goals of this western-blot were:

- (i) to confirm that tg650 brain homogenates contained PrP^{res}, a biochemical marker of prion infection and strain identity; and
- (ii) to enable comparisons with published data, which are often expressed in terms of brain homogenate dilution factors or the minimum detectable amount of PrP^{res}.

For these reasons, we chose to normalize visual signal intensity rather than load equal brain equivalents. While we acknowledge the reviewer's interest in further detailing these values, we feel that an extended discussion of loading differences may distract from the main message and conclusions of the figure.

12) Page 13, line 14: Change "> 5 logs" to "{greater than or equal to} 5 logs."

Done

13) Table 3: Include PrPres levels to better compare to the humanized mice experiments.

We now provide in Supplementary Figure 1B and in Table 3 the PrP^{res} data with the human sCJD / vCJD samples. A comparison of the minimum detectable amount of PrP^{res} from tg650 infected mice and patients is added to the discussion (page 18 lines 10 to 15).

14) Figure 2: In part A, what do the vertical dotted lines indicate? Presumably, these are the dilution ranges for AUC calculations in parts B and C. Consider specifying this in the legend. However, the dotted lines make this figure busy and hard to interpret. Consider communicating this in a more simplistic way. In part B, a 2-log dilution range is a very small window for comparison. Ideally, I would recommend repeating this with another VV1 patient sample so that no data are excluded in analysis. In part C, only 4 logs of dilutions are being compared. However, according to Table 3, 5 logs could be compared when VV1 is excluded.

As correctly noted, the vertical dotted lines in part A originally indicated the range over which AUC calculations were performed for the comparisons in part B and C. Following your suggestion, we have removed the vertical lines to reduce visual clutter and have instead added arrows to clearly indicate the segments of the curves used for statistical analyses.

Regarding the VV1 subtype, we fully agree that additional patient samples would strengthen the analysis, especially given the narrow 2-log amplification range observed. However, as previously discussed, VV1 is a rare subtype, and we did not have access to other suitable patient-derived samples for inclusion in this study.

As for the 4-log comparison range shown in part C, we appreciate the reviewer's observation that a 5-log comparison could theoretically be performed if VV1 data were excluded. However, the usable amplification ranges are not uniform across strains when expressed in terms of amplifiable particle concentration, which we used to normalize data across samples. This approach was chosen to account for differences in prion load between brain homogenates and to ensure substrate-relevant comparison. The amplifiable particle concentration was estimated using the Spearman-Kärber method (or a trimmed variant as needed), as described in the Material and Methods section. Using this method, we found that the amplification range varied considerably: for example, MV2 could not be amplified above 20,000 particles/mL, whereas strains like vCJD and MM2-c could not be amplified below 6 particles per mL. To ensure consistency and comparability across strains, we limited our analyses to the range of 6 to 6,000 particles/mL -a 4-log window for which all strains (except VV1) had

usable data. We hope this explanation clarifies our rationale and the modifications we made to improve the clarity of Figure 2.

15) Page 15, Line 18: Remove "clinical cases" after "vCJD" due to redundancy.
Done

16) Page 16, Line 4: Change "substrate" to "substrates."
Done

Reviewer #2 (Comments for the Author):

The paper states that:

- 1) The PrP E219K substrate amplifies all 6 sCJD strains and vCJD strain over a broad dilution range within a few hours
- 2) The discrimination of sCJD subtype from vCJD relies on lag time

I agree with 1) but have reservations about statement 2) related to how these data show that the lag time varies with dilution factor.

Although a measurement of PrP^{res} is described in the methods section, the section goes on to state that the RT-QuIC inocula were prepared as 10% dilutions - so there was no standardisation of seed mass between subtypes in the seeding volume.

We thank reviewer 2 for raising this important point. We agree that appropriate standardization is essential for meaningful comparisons of amplification kinetics between prion strains.

While PrP^{res} concentration was measured for each inoculum, we did not use it for standardization because PrP^{res} is not equivalent to PrP^{Sc}, and it does not reflect the actual number of seeding-competent (i.e., amplifiable) particles. As discussed in our response to Reviewer 1 (point 5), prion assemblies differ between strains in terms of size, size distribution, and structure, all of which influence their seeding activity and interaction with the RT-QuIC substrate. This estimate more accurately reflects the concentration of templating interfaces -the biologically relevant entities driving amplification.

Importantly, RT-QuIC outcomes are substrate-dependent. The same inoculum yields different amplification profiles depending on the substrate used (40). Therefore, comparing amplification kinetics based on the calculated concentration of amplifiable particles (rather than on PrP^{res} concentration or simple brain homogenate dilution factors) allows a more accurate and biologically meaningful assessment of strain-specific differences.

We believe this approach provides a robust framework for interpreting lag time variations and supports the conclusion that strain discrimination can be achieved—at least in part—based on these standardized kinetics parameters.

Line 1 on page 9 says "all experiments were performed in quintuplicate" however looking at the result figures I think this should say "sample replicates" and not "experiments" as I think the results as presented are too clean to be the average of 5 RT-QuIC experiments. If those figures as presented are indeed the average results of 5 separate RT-QuIC experiments, then I would like to see the raw data.

If these are replicates and not distinct experiments, then the results presented are from 1 dilution series on each subtype. I think the differences in lag times are more likely due to the seed mass in the inocula rather than any intrinsic property of the E219K PrP substrate. I suspect that if the seed mass in the inocula are standardised, the differences in lag time between subtypes will disappear.

We thank the reviewer for this important observation and the opportunity to clarify. You are correct that the wording in the original manuscript was inaccurate. The experiments were performed in

quadruplicate for all tg650-passaged strains and in octuplicate for negative controls—not in quintuplicate as originally stated. We have corrected this in the revised manuscript (page 9, lines 5 to 7), which now reads: “All reactions were performed in quadruplicate, except for tg650 negative controls which were run in octuplicate.” Each dilution of a given inoculum was used to seed four independent RT-QulC reactions (i.e., technical replicates). The resulting fluorescence curves were fitted using the equation described in the Materials and Methods (page 9, line 9), which allowed us to extract and compare kinetics parameters. This fitting process also contributes to the smooth appearance of the curves. As described in the manuscript (e.g., page 10 lines 13 to 14, page 13 lines 10 to 11, page 14 lines 10 to 11), we averaged only the positive amplification curves for each strain and each dilution, since combining positive and negative traces would not yield meaning estimates of amplification kinetics.

Regarding your request for data transparency, we provide you an excel file containing the raw data. The raw data can be published as supplementary materials or anywhere else if required.

As discussed previously, we did not standardize based on PrP^{res} mass, as this does not accurately reflect the amplification potential of the samples. For example, tg650-vCJD BH contained the highest amount of PrP^{res}—more than 20-fold greater than other strains—yet it exhibited the longest lag time in our assays. This illustrates that PrP^{res} content is not a reliable proxy for seeding activity. Differences in lag time between subtypes are not solely attributable to seed mass, but rather to a combination of strain-specific properties and substrate compatibility.

Other comments:

I am unclear how the models were used for fitting, how they were fitted or the value in using this approach over another. Although I haven't used nlme or lmPerm, the documentation for these packages reads like multiple models are available within these packages, so saying "we used this package" doesn't add value to the section and introduces more questions than answers.

We thank the reviewer for this comment and the opportunity to clarify our statistical approach. To address your concern, we revised the relevant sections of the manuscript to provide a clearer and more explicit description of the models used and the rationale for their selection.

In the Results section:

- Page 12, lines 8 to 11 now reads: “Statistical analyses were conducted using a linear model with a strain effect, applying both classical and permutation tests to assess the robustness of the results given the limited number of replicates. Both methods yielded similar results; Table 2 reports *P*-values obtained from the permutation test.”
- Page 13, lines 16 to 17 refer to the analysis mentioned on page 12: “we applied the same statistical analyses as described earlier to the AUC of lag time versus particle concentration”
- Page 15, lines 1 to 5 similarly states: “Statistical comparisons were conducted using a linear model with a strain effect, applying both classical and permutation tests to assess the robustness of the results given the limited number of replicates. Both methods yielded similar results. Table 4 reports the *P*-values obtained from the permutation test,”

In the Material and Methods section (page 9, lines 18–23), we now specify:

“For comparison of strains or patient isolates, the area under the curve of lag time (of RT-QulC amplification reactions) versus particle concentration was computed for each sample. Statistical significance was assessed using a linear model with a fixed strain effect applying both classical and permutation tests (with lmPerm package(51) in R) to assess the robustness of the results given the limited number of replicates. *P*-values were adjusted for multiple comparisons using the Tukey method.”

We hope this revised description provides greater transparency and clarity regarding the statistical methodology.

To further clarify, although the nlme package was originally mentioned in the text, we did not use it in the final analysis. All linear model fitting was done using the base R lm() function and the lmPerm package for permutation testing. We have removed references to nlme accordingly.

Table 2 - there is one p value in each box but the legend says 2 models were used to "evaluate differences" (in what?! presumably lag times? or goodness of fit?) so please give more details about this.

We agree that the original table legend required greater clarity regarding the statistical comparisons being performed.

Tables 2 and 4 present *P*-values derived from statistical analyses comparing AUC for lag time versus particle concentration across prion strains. Comparisons were conducted using a linear model with a strain effect, and both classical and permutation-based tests were applied. Only these *P*-values from permutation tests are presented since the ones from classical tests are similar and permutation tests are more robust than classical ones due to the small number of samples.

To address your comment, the text was modified (see above) and we revised the corresponding legends for clarity:

Table 2: "*P*-values from statistical analyses of the AUC of lag time as a function of particle concentration to compare tg650-passaged CJD prion strains. A linear model with permutation tests was used to evaluate differences between strains."

Table 4: "*P*-values from statistical analyses of the AUC of lag time as a function of particle concentration to compare human BHs infected with sCJD and vCJD prions. A linear model with permutation tests was used to evaluate differences between human prions. The first row for each pairwise comparison corresponds to analyses of the AUC over a 2-log dilution range; the second row corresponds to analyses of the AUC over a 4-log dilution range."

We hope these revisions provide the necessary details and improve the clarity of the data presentation.

There is a mixture of references to 40 hours and 50 hour duration in the manuscript, why the difference in endpoints between experiments/figures?

The difference in duration reflects to experimental contexts: amplifications using patient BHs were conducted over 40 hours, whereas amplifications using tg650-passaged prions were run for 50 hours. The longer duration for the tg650-passaged samples was necessary because certain strains - particularly vCJD and MM2-c - required additional time to reach a plateau in fluorescence. Truncating the experiments at 40 hours in these cases would have resulted in a loss of critical kinetic information and potentially underestimated amplification capacity.

p15 line 25 - Ok, no autopolymerisation is shown in the data, but many other RT-QuIC publications show stability of other PrP substrates up to 100 hrs. Why is being stable for 40 hrs such an advantage in this situation?

We thank the reviewer for this observation and fully agree that stability over 40 hours is not inherently advantageous when compared to substrates that remain stable for up to 100 hours. Our intention was not to claim that 40- or 50-hour stability is superior, but rather to emphasize that the absence of auto-polymerization with the duration of our experiments is critical for maintaining a clear distinction between positive and negative reactions. This is particularly relevant for diagnostic and comparative purposes, where premature substrate polymerization can obscure amplification signals.

We also aimed at comparing the auto-polymerization propensity of PrP E219K with other substrates. During this period, the PrP E219K substrate remained stable, without signs of spontaneous aggregation. In contrast, several other substrates described in the literature auto-polymerize earlier:

for example, hamster 90-231 at 15 hours (57), mouse PrP at 35 hours (55), and elk PrP at 45 hours (56), as cited in our manuscript page 17 lines 16 to 18.

p16 lines 17 -25. These data haven't been shown.

The data mentioned by Reviewer 2 are indeed included in the manuscript and are derived from the calculations based on values reported in Table 1. Specifically, we used these data to estimate the lowest detectable amount of PrP^{res} for each strain, in order to enable comparison with previous studies that present their detection limits in this format.

Fig 2. - Regarding the "mg (brain equivalent)" on the legend what does that mean - is that the amount of PrP^{res} in the loaded sample? or scaled up to the total sample? or the total brain? This is unclear as written. Was the intensity of all PrP^{res} bands in each lane used to determine this mass, or just one (which?) band in each lane? More detail as to how this was done would be useful.

PrP^{res} was purified from brain homogenates using a multistep procedure involving, dilution, protease digestion and centrifugation, as described in the Material and Methods. Based on the dilution factors and the final resuspension volume, we calculated the equivalent amount (in mg) of the starting brain tissue that corresponds to the sample volume loaded into each gel lane. Since the actual brain tissue no longer exists in its original form following purification, we used the term "brain equivalent" to refer to the calculated initial mass of brain material represented by the loaded sample. This is stated in the Material and Methods section, page 8 lines 16 to 18: "The amount of brain loaded per lane was expressed as "brain equivalent" (mg), *i.e.* the calculated mass of the original processed tissue in the sample volume."

To make this clearer to the reader, we have also updated the legend of Supplementary Figure 1 (formerly Supplementary Figure 2) to include the following sentence: "the amount of brain loaded per lane is expressed as "brain equivalent" (mg), *i.e.* the calculated mass of original processed tissue in the sample volume."

Regarding quantification:

PrP^C and PrP^{Sc} both contain two glycosylation sites, and PrP^{res} typically appears as three distinct bands on western-blot; the unglycosylated form (~19-21 kDa band, depending on the strain), and the mono- and bi-glycosylated forms at higher molecular weights. For quantification, we considered the combined intensity of all three glycoforms to determine PrP^{res} abundance. This analysis was performed using Image Lab software, calibrated against a standard curve derived from purified recombinant human PrP. We also updated the Material and Methods section accordingly (page 8, lines 13 to 15), which now states: "PrP^{res} size and abundance were determined by quantifying all three glycoforms using Image Lab software and a calibration curve generated from purified recombinant human PrP."

We hope this addresses the reviewer's concerns and clarifies the basis of the data presentation.

Fig 3. - Not all plots have SD overlaid - why? While the plots are colorful, they will be difficult for colorblind people to parse and are difficult to interpret as currently presented. I would suggest replotting with datapoints marked as e.g. squares, triangles, dotted lines, solid lines, etc. As currently presented, it's really hard to tell which dilutions are 10⁻³ and which are 10⁻⁹. I'm not sure whether all dilutions are present for each subtype. If not, why not?

You are correct that not all plots in Supplementary Figures 2 and 3 display SD overlays. As explained in the manuscript (page 10 lines 13 to 14; page 13 lines 10 to 11 and page 14 lines 10 to 11) and in the figure legends, we chose to present only the mean and SD of positive amplification reactions, excluding negative replicates to reduce clutter and improve interpretability. The only exception is the negative control, which is shown as a reference. Our rationale was to focus on meaningful amplification curves and to avoid averaging positive and negative traces, which would not accurately reflect amplification kinetics.

Regarding your suggestion to replot using shapes or line styles to aid interpretability, particularly for colorblind readers: we fully agree that accessibility is important. However, due to the large number of curves and closely spaced time points, replacing color with markers or grayscale lines made the figures far less readable. We explored several alternative plotting options—such as shape-coded data points and dashed lines—but these resulted in overly dense and visually confusing plots (an example is shown below).

Ultimately, we decided to retain the color version to maintain clarity and highlight dilution-dependent trends in amplification kinetics.

Concerning the presence of different dilutions per subtype: not all strains were successfully amplified at every dilution level. This variation reflects biological differences in amplification range, which is discussed in the Results and reflected in Tables 1 and 3. For example, VV1 amplified over only a 2-log range, whereas other strains reached up to 7 logs. We clarified this point in the relevant sections of the manuscript.

We hope these explanations justify our design choices and clarify the structure and content of the figures.

Typographical errors:

There are some errors and missing words.

P10 line 11 I think should read "Fig 1A-H"

We believe the current citation is accurate: it is specifically in Figure 1A that lag time is plotted as a function of particle concentration. Therefore, no correction was necessary at this location.

p11 line 24 "degree of discrimination"

Thank you, the text has been changed to "some discriminations".

p12 line 13 "began at 11 and 31 hrs"

We apologize, the word between was missing.

p12 line 16 missing letter - "lag"

This error has been corrected.

p15 line 24 "addresses" (present tense) not addressed

done

Re: Spectrum00292-25R1 (**Human PrP E219K: a new and promising substrate for robust RT-QuIC amplification of human prions with potential for strain discrimination**)

Dear Dr. Davy Martin:

Your manuscript has been accepted, and I am forwarding it to the ASM production staff for publication. Your paper will first be checked to make sure all elements meet the technical requirements. ASM staff will contact you if anything needs to be revised before copyediting and production can begin. Otherwise, you will be notified when your proofs are ready to be viewed.

Sincerely,
Isaac Solomon
Editor
Microbiology Spectrum

Reviewer #1 (Comments for the Author):

The authors effectively addressed prior concerns and have opted to use more precise language throughout based on their findings, particularly with their prior claims regarding strain discrimination and the range of amplification lag times between strains. Moreover the sensitivity of the assay between humanized mice and patient brain samples has been clearly described. Additionally, figures have been modified in-line with previous suggestions and effectively illustrate the novelty of using human PrP E219K as a substrate for RT-QuIC amplification and is an improvement over current approaches for CJD diagnostics.